# Crystal Structure of a Variant PAM2 Motif of LARP4B Bound to the MLLE Domain of PABPC1

**DOI:** 10.3390/biom10060872

**Published:** 2020-06-06

**Authors:** Clemens Grimm, Jann-Patrick Pelz, Cornelius Schneider, Katrin Schäffler, Utz Fischer

**Affiliations:** Department of Biochemistry, Theodor Boveri Institute, Biocenter of the University of Würzburg, 97070 Würzburg, Germany; jann-patrick.pelz@msd.de (J.-P.P.); cornelius.schneider@uni-wuerzburg.de (C.S.); katrin.schaeffler@uni-wuerzburg.de (K.S.); utz.fischer@biozentrum.uni-wuerzburg.de (U.F.)

**Keywords:** PAM2w, PAM2, PABC1, MLLE domain, PABP, Poly(A) binding protein

## Abstract

Eukaryotic cells determine the protein output of their genetic program by regulating mRNA transcription, localization, translation and turnover rates. This regulation is accomplished by an ensemble of RNA-binding proteins (RBPs) that bind to any given mRNA, thus forming mRNPs. Poly(A) binding proteins (PABPs) are prominent members of virtually all mRNPs that possess poly(A) tails. They serve as multifunctional scaffolds, allowing the recruitment of diverse factors containing a poly(A)-interacting motif (PAM) into mRNPs. We present the crystal structure of the variant PAM motif (termed PAM2w) in the N-terminal part of the positive translation factor LARP4B, which binds to the MLLE domain of the poly(A) binding protein C1 cytoplasmic 1 (PABPC1). The structural analysis, along with mutational studies in vitro and in vivo, uncovered a new mode of interaction between PAM2 motifs and MLLE domains.

## 1. Introduction

In higher eukaryotes, posttranscriptional mechanisms contribute significantly to gene expression. The mRNA, which is translated into proteins by the ribosomes, has first to undergo several maturation steps, from the primary pre-mRNA transcript to the mature mRNA. These include pre mRNA splicing, capping and polyadenylation, which are a prerequisite for the recruitment of additional factors, which contribute to the formation of the functional messenger ribonucleoprotein particle (mRNP). The vast majority of cellular mRNA is polyadenylated on its 3′-end, allowing binding of poly(A) binding proteins (PABPs). These proteins in turn serve as multifunctional scaffolds, enabling the recruitment of diverse factors to mRNPs [1,2]. PABPs typically contain four consecutive RNA recognition motifs (RRMs 1–4), the two N-terminal RNA recognition motif (RRMs) being necessary and sufficient for binding to the poly(A) tails of mRNAs [3,4,5,6]. The C-terminal part, in contrast, contains an unstructured, proline-rich region of low conservation [7,8,9,10,11] and a highly conserved domain termed MLLE (*Mademoiselle*), characterized by its invariant KITG (Lysine, Isoleucine, Threonine, Glycine) MLLE signature motif [12]. A large number of proteins interacting with PABP-MLLE have been identified, including several translation factors, such as eIF4G [13], Paip1 [14], Paip2 [15] and eRF3 [15]. These proteins contact PABP via a common motif of 12–15 amino acid residues, known as PABP interacting motif 2 (PAM2) [16,17].

Several structures of PAM2 motifs bound to MLLE domains have recently been determined [15,18,19,20,21,22]; see [17] for a review. Within those structures, the respective MLLE domain appears as a bundle of four or five α-helices. Two pockets formed in between helix 2 and 3, as well as helix 3 and 5, respectively, are major determinants for the PAM2 interaction.

The La-related protein (LARP) family comprises five major groups [23,24,25], whose common structural hallmark is the La module, a combination of the La motif and an adjacent RRM. Structural [26,27,28] and biochemical studies on the genuine La protein revealed the specific binding of the La module to 3′ oligo-uridylic acid stretches [29,30]. Accordingly, La associates predominantly with polymerase III transcripts in vivo, and influences their stability and/or maturation pathways [31]. LARP7 family members likewise bind the 3′end of Pol III transcripts. However, they have apparently developed specificity for a subset of Pol III RNA targets [32,33,34], and facilitate specific functions, such as the stabilization of 7SKRNA or the 2′O methylation of the spliceosomal U6 small nuclear ribonucleic acid (snRNA) [35,36,37,38,39]

In contrast, other LARP members, including 1, 4 and 6, interact with mRNA rather than with Pol III transcripts [40,41,42,43]. Moreover, with the exception of LARP6, these cytoplasmic LARP proteins all harbour PAM2 motifs, and likely bind PABPC1 [44,45,46,47]. Within this group, LARP4 and LARP4B stand apart, as they harbor a variant PAM2 motif close to their N-termini (termed PAM2w), in which a conserved Tyrosine residue is replaced by Tryptophan [48]. The crystal structure of the LARP4 PAM2w motif, bound to the PABC MLLE domain [49], shows that the PAM2w Tryptophan residue binds in the same pocket on the MLLE surface that harbors the paralogous Tyrosine residue, as seen in structures of MLLE with the classical PAM2 motif.

Here we describe the structural analysis of the interaction of LARP4B with the MLLE domain of PABC. We have shown previously that LARP4B is associated with the ribosome-associated receptor for activated C kinase 1 (RACK1), as well as PABPC1 [47], and that it might act as a stimulatory factor in translation [43]. More recent evidence suggests that the observed increase in translational output upon overexpression of LARP4B can be attributed, at least in part, to an increase in Poly(A) tail length and a stabilization of the transcripts (Mattijssen et al. 2017).

We show that the LARP4B PAM2w motif binds MLLE in the canonical binding cleft, albeit with a divergent binding mode, with respect to the conformation of the PAM2w Tryptophane residue.

## 2. Materials and Methods

### 2.1. MLLE Protein Expression, Purification and Complex Crystallization

The MLLE domain (amino acids 543-621) of human PABPC1 was cloned into vector pETM-30, and expressed in *Escherichia coli* strain BL21 (DE3) pLysS Rosetta2. The LARP4B–PAM2w peptide was custom-synthesized and purchased from PSL GmbH (Heidelberg, Germany). Crystals were grown with the hanging drop vapor diffusion method by mixing 3 µL of MLLE/peptide solution (100 mM sodium chloride, 2mM ß-mercaptoethanol, 10 mM HEPES, pH 7.5 at a concentration of 43 g/L supplemented with LARP4B–PAM2w peptide in a 1:1.5 molar ratio) with 3 µL of reservoir solution (1.5 M magnesium sulfate, 0.1 M Bis-Tris, pH 6.5).

### 2.2. Crystallographic Methods

A diffraction dataset for a single crystal was collected at beamline ID14-4 of the European Synchrotron Radiation Facility (ESRF). Data processing and scaling were carried out using the program XDS [50]. The structure was solved by molecular replacement with the program PHASER [51], using the coordinates of unliganded MLLE [15] (PDB entry 3KUR) as a search model. After manual building of the peptide, the model was refined with REFMAC5 [52]. Model coordinates and the diffraction dataset were deposited within the Protein Data Bank (PDB) as entry 3PTH. Data collection and refinement statistics are given in Table 1.

### 2.3. Cloning, Mutagenesis and Antibodies

For in vitro translation of [^35^S]-labelled proteins, the respective cDNAs were cloned into expression vector pHA (Invitrogen, Darmstadt, Germany). The luciferase expression vectors pGL3 and pRL-TK were purchased from Promega (Madison, USA) [53]. In vitro translated proteins were produced using the TnT-T7 quick coupled transcription/translation system (Promega, Madison, WI, USA). Site-directed mutagenesis was performed with the Quickchange Site-Directed Mutagenesis Kit (Stratagene, Kirkland, WA, USA) according to the manufacturer’s instructions. Antibodies used were obtained as described previously [47].

### 2.4. Cell Culture and Luciferase Assay

HEK293 as well as HeLa cells were grown in DMEM containing 10% (*v*/*v*) FCS and penicillin/streptomycin. To analyze translation in vivo, LARP4B or mutant LARP4B, each cloned into vector pHA, were co-transfected with luciferase expression vectors (pGL3 and pRL-TK) into HEK293 cells using Nanofectin (PAA, Ontario, CA, USA). 48 h post transfection, cell extracts were prepared [47] and luciferase activity was determined using a standard chemoluminescence detection procedure together with a dual luciferase reporter assay system according to the manufacturer’s instructions (Promega, Madison, USA).

### 2.5. Immunological Procedures and In Vitro Binding Studies

Immunoprecipitations and the in vitro binding assay were performed as described previously [37,47]. Proteins were separated by sodium dodecyl sulfate polyacrylamide gel electrophoresis (SDS-PAGE) and visualized by Western blotting [54] or analysed by autoradiography [55], respectively. Immunocytochemistry and stress treatment were carried out as described [56].

## 3. Results

### 3.1. LARP4B Contains a Variant PAM2 Motif Necessary for Efficient PABC1 Binding

Our previous in vitro binding experiments, with recombinant LARP4B, demonstrated a direct interaction of PABPC1 with LARP4B [47]. On the LARP4B side, two modules in its N-terminus (amino acids 1–153) and C-terminus (amino acids 308–738) were defined as independent binding modules (Figure 1A). This prompted us to examine the corresponding sequences for possible interaction motifs. Via multiple sequence alignment and a motif search, we discovered a sequence stretch near the N-terminus of LARP4B (residues 56 to 63) that was closely related to the PAM2 motif (Figure 1B). However, the motif found in LARP4B differs from the canonical PAM2 consensus sequence LNxxAxEFxP [17], in that it contains a tryptophan, instead of an otherwise highly conserved phenylalanine residue, at position 10 of the consensus sequence. The phenylalanine residue in this position is a main determinant of the PAM2/MLLE interaction, as can be predicted from all complex structures known so far. The sequence in LARP4B represents a variant PAM2 motif, termed PAM2w [48,49], that can interact with the MLLE domain of PABC1.

To test this for LARP4B, we co-translated and [^35^S]-methionine-labeled PABPC1, along with either a wild-type LARP4B protein or a mutant thereof, in which the variant tryptophan (position 63 in the human protein) had been replaced by lysine. We predicted that this substitution would interfere with the binding of LARP4B to MLLE, as a corresponding substitution in canonical PAM2 motifs is expected to be sterically incompatible with this interaction. The co-translated proteins were co-immunoprecipitated with antibodies against LARP4B, and analyzed by autoradiography (Figure 1C). Whereas the wild-type protein formed a stoichiometric complex with PABPC1, binding to the mutant version was markedly reduced (compare lanes 1 and 3). No proteins were immunoprecipitated by a control serum, illustrating the specificity of the antibody used (lanes 2 and 4). Next, we tested whether the same effect was observable in vivo. Plasmids encoding the wild-type and mutant LARP4B protein were co-transfected, with PABPC1-encoding plasmids, into HEK293 cells. After this, lysis immunoprecipitations were carried out as described above, and the proteins were detected by Western blotting. Mutant LARP4B bound slightly more weakly to PABPC1, as compared to the wild type. However, this effect seemed to not be as strong as in vitro. These results gave us a first hint that the PAM2w domain might contribute to the interaction of LARP4B with PABPC1 in vitro, but that in vivo additional factors or other regions in LARP4B could likely influence binding.

### 3.2. Crystal Structure of the LARP4B PAM2w/MLLE Complex

In order to understand the molecular details of the uncovered interaction, we co-crystallized a peptide representing the LARP4B PAM2w motif (residues 56 to 70), together with the PABC1 MLLE domain (residues 543 to 621), and solved the crystal structure. Figure 2 shows the molecular details of the peptide/protein interaction.

Within the crystal structure, a single LARP4B–PAM2w complex contacts a neighbouring, symmetry-related MLLE entity, providing an extended interface that creates a deep cleft in the interface region into which the peptide binds, running over or in between the surfaces of both symmetry-related MLLE molecules (Figure 2). Binding of the N-terminal, the canonical part of the LARP4B–PAM2w peptide, is concordant with the structures reported for other PAM2 motifs bound to MLLE (see text below and also Figure 3). A hydrophobic binding pocket (A) in between helix α2 and α3, lined by residues Q560, E564, L566, F567, K580, T582 and L586, harbours W63 of the LARP4B–PAM2w motif (Figure 2). This demonstrates the ability of pocket A to either bind a tryptophan, or alternatively, a less bulky phenylalanine residue. This behavior conforms to the MLLE conserved binding sequence pattern reported previously [17]. A second hydrophobic pocket (B) lies between helix α3 and α5, outlined by residues L569, T576, L597, P600 and L603, harbouring L56 of the PAM2w motif. In between peptide residues L56 and W63, a β-turn is stabilized by P58. While A60 is accommodated by a shallow pocket on the surface of MLLE molecule A [crystallographic symmetry operation (x, y, z)], E55, N57, N59 and E61, the remaining hydrophilic residues of the canonical PAM2w motif, point away from the MLLE A surface into the solvent. The C-terminal, a variable part of the LARP4B–PAM2w peptide, is located within the gap between the contacting symmetry-related MLLE molecules B [symmetry operation (x − 1, y, z)] and C [symmetry operation (x − 0.5, −y + 0.5, −z)]. To enter this gap, the peptide performs an almost 90° turn around G64. Inside, V67 binds into a hydrophobic pocket located on the MLLE B surface, interacting mainly with L569 and P600. The neighbouring residue, Leu68, binds into a hydrophobic pocket on MLLE C, outlined by L577, I581, V611 and L614. Well-defined electron density was observed for the whole peptide, and only the C-terminal residue, L70, was disordered and therefore omitted from the model.

Altogether, a total continuous surface of 1621 Å^2^, or 33.3% of a single MLLE molecule’s accessible surface area, is buried within the MLLE–MLLE and MLLE–peptide contacts. The result of these interactions is an extended and rugged interface, centred about the LARP4B–PAM2w peptide. This is reflected in a zone of comparably low B-factors within this region.

### 3.3. PAM2w/PAM2 Sequences Can Be Divided into a Canonical and a Variable Part

A superposition of the MLLE co-structures, with PAM2 or PAM2w motifs from LARP4B, LARP4, Ataxin2, eRF3 (N-terminal as well as C-terminal PAM2 motif) and Paip2, clearly illustrates the bipartite character of the PAM2–MLLE interactions (Figure 3). The C-terminal, the canonical parts of the bound PAM2 peptides, overlap closely, exhibiting the same mode of binding to the MLLE surface within all structures. The main anchoring points for the binding of the canonical motif parts are, in all cases, the two hydrophobic pockets A and B. In between those, all peptides display a β-turn, which is stabilized by a hydrogen bond between the conserved PAM2–asparagine L**N**xxAxx[F/W] (position 4 of the consensus sequence, corresponding to Asn57 of LARP4B) preceding the turn, and the backboncensus sequence (LNx**x**Axx[F/W]). The conserved alanine residue LNxx**A**xx[F/W] following the turn points towards the MLLE surface, and therefore a bulkier side chain at this position would sterically be highly unfavourable. In the absence of a significant induced fit, the relatively wide and shallow hydrophobic pockets A and B are able to harbor different hydrophobic side chains. Judging from the frequency of occurrence within so-far identified PAM2 motifs, pocket B has a preference for Phe, but is also able to harbor a Trp residue, as shown previously for LARP4 by [49], as well as in this study. Of note, the versatility of this pocket is also reflected by its ability to harbor the bound tryptophane in different sidechain conformations [compare LARP4(Trp22) and LARP4B(Trp63) in Figure 3]. Correspondingly, pocket A seems to prefer Leu, which binds in a radial mode with respect to the protein surface. However, the C-terminal of one of the two overlapping eRF3 PAM2 motifs shows an F residue binding with a tangential mode into pocket A. While the variable sequence downstream of the canonical motif contributes significantly to the PAM2/MLLE interaction, this region of the corresponding bound peptides runs over completely different parts of the MLLE surface within the different co-structures. Likewise, the GW182 DUF peptide is anchored by its F1389 residue within pocket B, and locally shares a few more interactions in the adjacent regions with PAM2 peptides. However, the rest of the contacts differ largely from those seen for canonical PAM2 sites.

### 3.4. Disruption of the PAM2w/MLLE Interaction Does Not Affect Recruitment of LARP4B to Stress Granules upon Arsenite Stress

We tested whether the integrity of the PAM2w motif is critical for the function of LARP4B. We showed previously that LARP4B translocates to subcellular domains, termed stress granules (SGs), under various stress stimuli [47]; [57]. It is believed that SGs under these conditions serve as storage pools for stalled, mRNA-bound translation initiation complexes formed upon polyribosome disassembly [57,58]. As the PAM2w–MLLE interaction of LARP4B with PABPC1 appeared to have only minor effects on LARB4B binding to PABPC1 in vivo, we set out to test whether the LARP4B would be recruited to the same mRNPs in vivo, upon disruption of the PAM2w–PABPC1 interaction. To this end we investigated whether the LARP4B_W(63)→K_ mutant recapitulates our previously observed recruitment of *wild type* (wt) LARP4B to stress granules upon arsenite treatment [47].

We therefore analyzed the intracellular localization of a HA-tagged LARP4B truncation (the N-terminal part of the protein, including the PAM2w motif), as well as the LARP4B_W(63)→K_ mutant under normal growth and under stress conditions (Figure 4 shows a schematic model of the LARP4B constructs used). Under normal conditions, the LARP4B variants were homogenously distributed in the cytoplasm (Figure 4, panels A–E, K–O and U–Y). As shown in Figure 4, all tested proteins translocated upon stress induction with arsenite to SGs (compare panels F–J, P–T and Z–D’). The Fragile X Mental Retardation Protein (FMRP), a well-established SG marker protein, served as a control in these experiments (see Figure 4, panels B, G, L, Q, V and A’). In sum, these experiments indicate that uncovered interaction, while strongly affecting LARP4B’s interaction with PABPC1 in vitro, seems to not be sufficient in vivo to disrupt the recruitment of LARP4B to stress granules. These results suggest that additional factors, until now unknown, contribute to the recruitment of LARP4B to its native mRNPs.

## 4. Discussion

A large variety of different proteins are recruited to mRNPs by virtue of their PAM motifs. Accordingly, this interaction has been studied intensely, both at the biochemical and structural level. Here, we have uncovered the atomic details of the LARP4B–PAM2w interaction with the MLLE domain. A comparison of all relevant atomic structures revealed that the major determinants for the binding of PAM2w and PAM2 motifs reside within their canonical part, i.e., in the parts that are shared between both motifs. Therefore, binding of different proteins containing PAM2 or PAM2w motifs is likely to occur mutually exclusively. Given the large number of protein factors containing a PAM2/PAM2w or related motif, they are expected to compete for binding to their respective binding sites located on the PABP–MLLE domain. In this regard, the variable part of PAM2/PAM2w sites, and/or other binding surfaces located in other parts of the protein, might help to fine tune and diversify these interactions, while the canonical part might represent a common ‘interaction module’, providing a basal affinity. At the border of both motif parts, the LNxxAxx[**F/W**] F/W residue could provide a switch point, sending the variable part of the bound peptide chain to different MLLE surface areas, depending on the different stereo-chemistries imposed by either a F or a W residue.

In this study, we observe the binding of the variable part of the PAM2w motif, extruding from the canonical MLLE complex, into a cleft on the surface of a MLLE unit neighbouring in the crystal packing. In this regard it is worth mentioning that the MLLE domain used here failed to form crystals on its own, but rapidly crystallized upon addition of the PAM2w peptide. This can be explained by the stable and extended contacts that emerge from the peptide contacting three distinct MLLE molecules. There is currently no data that supports this behavior in vitro, in solution. Nevertheless, one could hypothesize that, in vivo, the PAM2w site, located within the largely unstructured and flexible N-terminus of LARP4B, could simultaneously contact two or more distinct MLLE units of neighbouring, poly(A)-bound PABC molecules. However, it should not go unnoticed that the cross-species sequence conservation is only high in the canonical part of the LARP4B–PAM2w motif, and comparably low in its variable part [59].

Biochemical studies have suggested that LARP4B is recruited by, and stably bound on, mRNPs by virtue of an elaborate protein and mRNA network [47]. It was hence not entirely unexpected that a missense mutation in PAM2w, or even the entire deletion of the motif, failed to interfere with the recruitment of LARP4B to this mRNP. We speculate that the interface between PAM2w and MLLE contributes only partially to the stable recruitment of LARP4B to mRNPs. Most likely, recruitment of LARP4B to its mRNPs is aided by the C-terminal region, which was shown to co-immunoprecipitate with PABPC1 independently of PAM2w [47]. Alternatively, and not mutually exclusively, our studies may indicate an additional function of LARP4B that critically depends on the intact PAM2w motif. Recently, LARP4B overexpression was shown to increase Poly(A) tail length and increase mRNA stability (Mattijssen et al. 2017). It is tempting to speculate that the disruption of the LARP4B–PAM2w and –MLLE interaction has an influence on this important role of LARP4B, similar to what was shown for the related, but functionally different, LARP4 (Mattijssen et al. 2017).

In addition, it was reported that LARP4 possesses a PAM2 variant similar to that of LARP4B, and the authors of this study presented a crystal structure of the corresponding PAM2w–MLLE complex [49]. This structure features a similar binding mode for the canonical motif part. However, this structure differed from the one we have reported in this manuscript, in that the LARP4–PAM2w peptide bound to one MLLE unit only. This raises the interesting possibility that different PAM2 variants might use alternative modes of interaction.

## Figures and Tables

**Figure 1 biomolecules-10-00872-f001:**
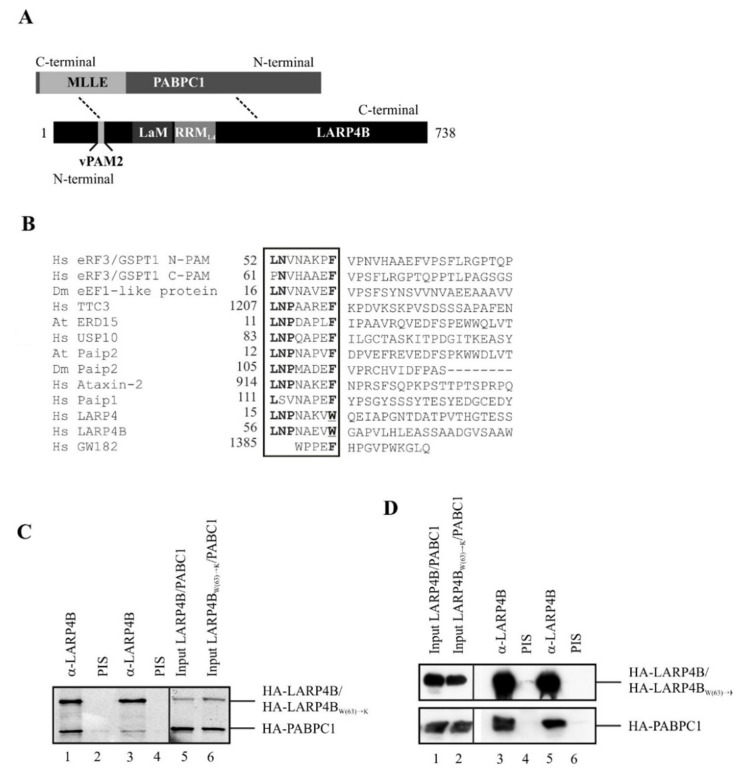
Characterisation of the PAM2w motif and its binding to poly(A) binding protein C1 cytoplasmic 1 Mademoiselle domain (PABPC1-MLLE). (**A**) A schematic model of the LARP4B–PABPC1 interaction. (**B**) Sequence alignment of PAM2 motifs of several proteins from different organisms (Hs—Homo sapiens; Dm—Drosophila melanogaster; At—Arabidopsis thaliana): Hs eRF3 (elongation release factor 3), Hs TTC3 (E3 ubiquitin-ligase TTC3), Hs Ataxin-2, Paip1 [Poly(A) interacting protein 1], Hs LARP4 and LARP4B (La-related proteins 4 and 4B), Dm eEF1-like protein (eukaryotic elongation factor 1-like protein), Dm Paip2 [Poly(A) interacting protein 2], At. ERD15 (early response to dehydration 15), At. Paip2 [Poly(A) interacting protein 2]. Conserved amino acids are highlighted by bold characters, the variant tryptophan residues of LARP4 and LARP4B are underlined. (**C**) In vitro binding experiment using protein-G-Sepharose coupled antibody against LARP4B (lanes 1 and 3) or a pre-immune serum (lanes 2 and 4). HA-PABPC1, as well as HA-LARP4B or HA-LARP4B_(W63)→K_, respectively, were co-translated in vitro, [^35^S]-labeled and immunoprecipitated with the indicated antibodies. Bound proteins were detected by autoradiography (HA-LARP4B lanes 1-2, HA-LARP4B_(W63)→K_ lanes 3–4). Input, 5% of [^35^S]-labeled HA-PABPC1, HA-LARP4B (lane 5) or HA-LARP4B_(W63)→K_ (lane 6). (**D**) Immunoprecipitation of HEK293 cell extracts overexpressing either HA-LARP4B or HA-LARP4B_W(63)→K_ using an antibody against LARP4B (lanes 3 and 5) or a pre-immune serum (lanes 2 and 4). Lanes 1 and 2 show the transfection controls. The gel was analyzed by Western blotting.

**Figure 2 biomolecules-10-00872-f002:**
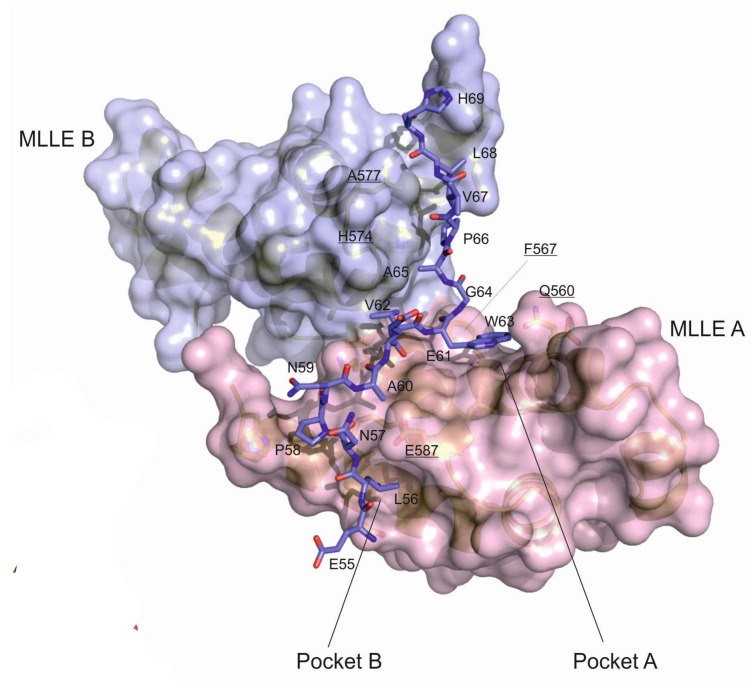
Surface representation of MLLE molecule A (pink) and B (blue) with the bound LARP4B–PAM2w peptide, shown as a stick model. MLLE residue names are underlined. (blue) and C (green).

**Figure 3 biomolecules-10-00872-f003:**
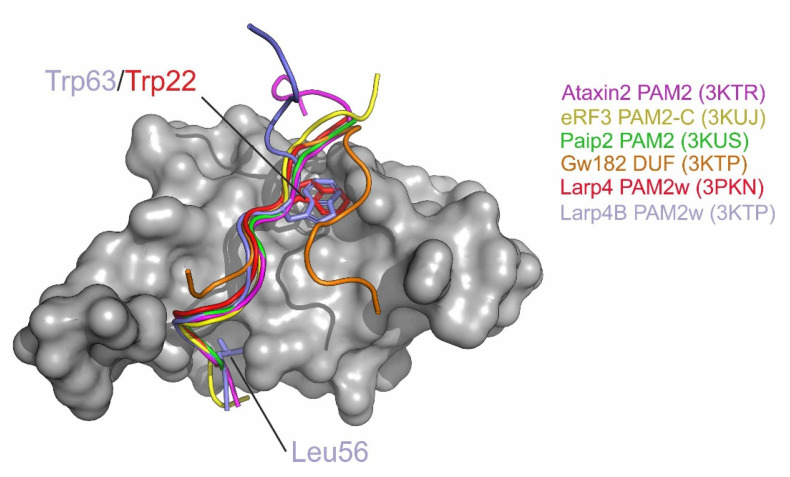
Superposition of the Cα-trace of LARP4B–PAM2w (blue, sidechains L56 and W63 shown) with Cα-traces of the LARP4–PAM2w (PDB ID 3PKN, red, sidechain W22 shown) and PAM2 peptides from PABP interacting protein (Paip2, PDB ID 3KUS, green), release factor 3 (eRF3 C-terminal motif, PDB ID 3KUJ, yellow) and Ataxin2 (PDB ID 3KTR, pink). The GW182 DUF peptide (3KTP) is depicted in orange.

**Figure 4 biomolecules-10-00872-f004:**
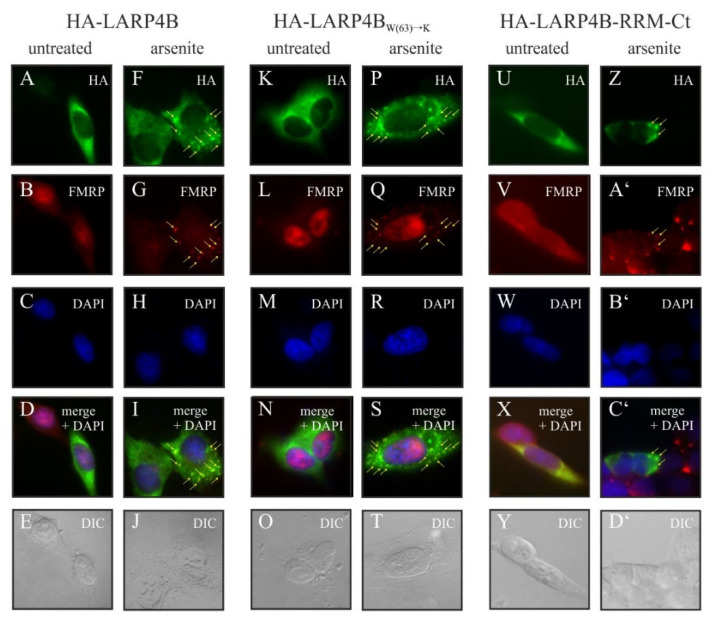
The C-terminus of LARP4B is sufficient to accumulate in stress granules. Upper part, depiction of the LARP4B constructs used for immunofluorescence. Lower part, immunofluorescence studies in HeLa cells transfected with the introduced constructs, using antibodies against Human influenza hemagglutinin (HA) and fragile X mental retardation protein (FMRP) as a stress granules marker protein. Cells were either mock-treated (panels **A**–**E**, **K**–**O**, **U**–**Y**) or treated with arsenite (panels **F**–**J**, **P**–**T** and **Z**–**D’**). SG are marked by arrows.

**Table 1 biomolecules-10-00872-t001:** Crystallographic data collection and refinement statistics.

Data Collection	
Space group	P 2_1_ 2_1_ 2_1_
Cell dimensions a, b, c [Å]	29.15, 57.19, 59.51
Resolution [Å] (outer shell)	41.2–1.7 (1.8–1.7)
I/sigma(I) (outer shell)	14.4 (2.4)
Completeness [%] (outer shell)	96.9 (86.8)
Redundancy (outer shell)	3.2 (2.4)
Wilson B-factor [Å^2^]	26.1
**Refinement**	
Resolution [Å]	41.2–1.7
No. of reflections	10805
R_work_/ R_free_	17.2/21.8
No. atoms	769
Water molecules	82
*B-factors [Å^2^]*	
overall	22.2
MLLE (Chain A)	19.2
Peptide (chain B)	24.0
Solvent (chain Z)	40.3
*RMS (**root-mean-square deviation*) *deviations from ideal*	
Bond lengths [Å]	0.017
Bond angles (°)	1.6
*Ramachandran statistics, residues in [%]*	
Most favoured regions	96.9
Additionally allowed regions	3.1
Generously allowed regions	0

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
