# Peer review of "Crystal Structure of a Variant PAM2 Motif of LARP4B Bound to the MLLE Domain of PABPC1"

_biomolecules, 2020, doi:10.3390/biom10060872_

Round 1

Reviewer 1 Report

The manuscript describes a crystal structure of a human variant PAM2 peptide from LARP4B and 2 MLLE domains of PABPC1 that was deposited in the data base a decade ago.  That is OK.  The structure presented in this manuscript has been available in the public domain PDB (3PTH) and has been reviewed in present day context in Dock-Bregeon et al., see figure 9 in RNA Biol, 1-16 (2019). Neveretheless, I would recommend publication of the present manuscript if it could be revised and made appropriate for Biomolecules and in a way that its impact would be improved by addressing the flaws noted below. Just because the authors describe a structure from a decade ago doesn't mean that they should resubmit a paper that was mostly written at that time, as this paper is distastefully presented.  The authors should re-write the manuscript in the here-and-now, not just inserting citations of current literature, but incorporating the science into proper context and interpretation of the results. It would appear as if the authors mistakenly uploaded a manuscript written ten years ago that was somehow updated with new citations.

major:

1) The paper describes the PAM2 sequence whose major feature as variant is that it contains a Trp instead of Phe at a very important position that is known to be a strong determinant of binding affinity in one of two hydrophobic pockets of the MLLE domain of PABC1. Yet they don't acknowledge the relevance of this unique substitution referred to as PAM2w in LARP4A since it was first described (ref 58) and attributed to that protein's activities (Cruz-Gallardo et al. Nucleic Acids Res 47, 4272-91, 2019, Mattijssen et al., Elife 6: e28889, 2017).

2) The authors must address the fact that they are not measuring translation in figure 4A, as they claim, they are just measuring luciferase output. The authors buried the citation of Mattijsen et al. ELife 2017 among other citations but do not refer to its important findings that are very relevant to theirs. It was shown in Mattijsen et al that LARP4A (aka LARP4) and LARP4B can increase the length of poly(A) tails of reporter mRNAs and endogenous mRNAs, and the stability and levels of the mRNAs. Therefore, it is likely that LARP4B increases the levels of the co-transfected luciferase reporter mRNAs. Thus, to claim that LARP4B has an effect on translation, they would have to divide the luciferase activity by the luciferase mRNA levels.

      Furthermore, Mattijsen et al show that LARP4B accumulates to very high levels the PAM2w contributes to poly(A) lengthening.  

Thus, this analysis is not meaningful as it is. Without showing mRNA levels or explicitly acknowledging that they are likely elevated due to LARP4 effects, the luciferase data should not be part of this manuscript nor referred to.

      This reviewer doesn't believe that the PAM2w of LARP4B is nonfunctional based on the data presented in this paper.

Why are there no error bars on the left graph? There are also no labels on the western blot and the empty control vector is not shown here. Lane 215: “able to drastically stimulate translation of both reporters.” There is not much of an increase in the Firefly signal, also there are no error bars, so we don’t know anything about the statistical significance.

3) The authors should not refer to old PAM2 consensus sequences and use old numbering when referring to their sequence.  Although they cite Xie et al, (ref 23) which illustrates an updated consensus, they don't use it.  They erroneously refer to the conserved phenylalanine as position 8, replaced by tryptophan because they haven't rewritten this old manuscript to bring it up to date; in modern time nomenclature consensus, this is position 10 (see Xie et al, ref 23)

4) The authors argue in the discussion that the structure that shows the downstream sequence beyond the PAM2 that engages the second MLLE domain in a noncanonical way may be meaningful but don't present any data to support that interpretation.  But nor do they discuss the data that argues against it, including from their data in fig 1C in which they concluded (line 131): "the wild-type protein formed a stoichiometric complex with PABPC1". 

A recent paper by Deragon shows alignment of the PAM2w sequences of multiple mammalian and other vertebrate LARP4B proteins (Deragon RNA Biology 2020, supplementary figure S3D). https://www.tandfonline.com/doi/full/10.1080/15476286.2020.1739930  The authors should consider/address that sequence homology is very high in the PAM2w sequence but is lower beyond it.  

Thus, in discussion they should also consider that the second MLLE in the crystal is an artifact and discuss experiments using LARP4B constructs mutagenized at residues downstream of the PAM2w that are observed to be in contact with the second MLLE. 

Finally, when first described in Results (line 150) the authors write: "Within the crystal structure, a single LARP4B-vPAM2 peptide and three neighboring MLLE molecules contact each other, producing an extended interface (Fig. 2B)." but there are only two MLLE domains shown in the figure 2 (there is no Fig. 2B).

5) Figure 4B: the IF pictures are not high quality, especially of the C-terminal fragment. If they have images that show more cells they should use those.  There are only 3 stress granules in the cells that overexpress the C-terminal fragment.

      A conceptual problem is that as presented, the authors argue that presence in stress granules, reflects translation, but this is incorrect.  According to current understanding, stress granules is where mRNAs reside that are not actively translated. 

      The negative data here don't add anything to this paper, similar to the luciferase data. 

Specific points:

  1. Figure 1D: Two bands are present for PABPC1 in both input lanes and lane 3 corresponding to IP with WT LARP4B, but only the upper band was IP with LARP4B-W63K. The authors should comment on this.

      How were the results quantified? Was this done with ECL or fluorescent antibodies? Was the HA-PABPC1 signal divided by the HA-LARP4B signal?

  1. Figure 1C and D: Please indicate in the figures which lane is WT LARP4B and which lane is mutant LARP4B.
  2. Figure 1C: How were the results quantified? Was the HA-PABPC1 signal divided by the HA-LARP4B signal?
  3. Page 6, line158: “like in all other PAM2-MLLE complex structures reported so far.” This same observation is already reported for LARP4 PAM2w
  4. Page 6, line 159: “Therefore, the generic pattern for the canonical PAM2 motif LNxxAxxF has to be extended to LNxxAxx[F/W]. “ Even though the authors cite Xie et al (2014), in which an updated consensus and logo is published that includes the F/W, they completely neglect to mention this.
  5. Line 195: "Thus, the interaction of vPAM2 with MLLE is not essential for LARP4B’s stimulatory role in translation." This must be deleted because they didn't measure translation, not by luciferase output as above and not by stress granules where translation doesn't occur
  6. Line 232: “As the vPAM2-MLLE interaction of LARP4B with PABPC1 appeared to be non-essential for LARP4Bs effect on translation, we assumed, that a truncation of LARP4B lacking the N-terminal part of the protein (including the vPAM2 motif) should be sufficient to accumulate in stress granules." This rationale doesn't make sense, it must be deleted , see above
  7. Line 251: “A comparison of all relevant atomic structures revealed that the major determinants or binding of vPAM2 and PAM2 motifs reside within their canonical part, i.e. in the parts that are shared between both motifs.” not all relevant atomic structures were used. What about LARP4?

Minor:

  • In their description of fig 1 they show that mutation of the Trp (W) decreases binding to PABPC1 in vitro and less so in vivo by IP. They note (line 139): "These results suggest that the vPAM2 domain contributes to the interaction of LARP4B to PABPC1 in vitro and in vivo although in the latter case additional factors are likely to influence binding (see also discussion)." First, they do not address this in the discussion but more importantly, they should refer to the other region depicted in figure 1a rather than "additional factors".
  • Page 2, line 44” Moreover these cytoplasmic LARP proteins all harbor PAM2 motifs and likely bind PABPC1 [45-48]
  • human LARP6 does not have PAM2
  • Page 3, line 90: delete: , respectively
  • lines 151, 157: refer to fig 2A, 2B but there is only fig. 2.

Author Response

Reviewer 1:

Comments and Suggestions for Authors
The manuscript describes a crystal structure of a human variant PAM2 peptide from LARP4B and 2 MLLE domains of PABPC1 that was deposited in the data base a decade ago.  That is OK.  The structure presented in this manuscript has been available in the public domain PDB (3PTH) and has been reviewed in present day context in Dock-Bregeon et al., see figure 9 in RNA Biol, 1-16 (2019). Neveretheless, I would recommend publication of the present manuscript if it could be revised and made appropriate for Biomolecules and in a way that its impact would be improved by addressing the flaws noted below. Just because the authors describe a structure from a decade ago doesn't mean that they should resubmit a paper that was mostly written at that time, as this paper is distastefully presented.  The authors should re-write the manuscript in the here-and-now, not just inserting citations of current literature, but incorporating the science into proper context and interpretation of the results. It would appear as if the authors mistakenly uploaded a manuscript written ten years ago that was somehow updated with new citations.

major:

1) The paper describes the PAM2 sequence whose major feature as variant is that it contains a Trp instead of Phe at a very important position that is known to be a strong determinant of binding affinity in one of two hydrophobic pockets of the MLLE domain of PABC1. Yet they don't acknowledge the relevance of this unique substitution referred to as PAM2w in LARP4A since it was first described (ref 58) and attributed to that protein's activities (Cruz-Gallardo et al. Nucleic Acids Res 47, 4272-91, 2019, Mattijssen et al., Elife 6: e28889, 2017).

We have modified the manuscript to include a clear reference to LARP4A and the appropriate literature. We have changed the nomenclature for the variant PAM2 motif to the currently used and generally accepted term “PAM2”. 

2) The authors must address the fact that they are not measuring translation in figure 4A, as they claim, they are just measuring luciferase output. The authors buried the citation of Mattijsen et al. ELife 2017 among other citations but do not refer to its important findings that are very relevant to theirs. It was shown in Mattijsen et al that LARP4A (aka LARP4) and LARP4B can increase the length of poly(A) tails of reporter mRNAs and endogenous mRNAs, and the stability and levels of the mRNAs. Therefore, it is likely that LARP4B increases the levels of the co-transfected luciferase reporter mRNAs. Thus, to claim that LARP4B has an effect on translation, they would have to divide the luciferase activity by the luciferase mRNA levels.

      Furthermore, Mattijsen et al show that LARP4B accumulates to very high levels the PAM2w contributes to poly(A) lengthening.

Thus, this analysis is not meaningful as it is. Without showing mRNA levels or explicitly acknowledging that they are likely elevated due to LARP4 effects, the luciferase data should not be part of this manuscript nor referred to.

      This reviewer doesn't believe that the PAM2w of LARP4B is nonfunctional based on the data presented in this paper.

Why are there no error bars on the left graph? There are also no labels on the western blot and the empty control vector is not shown here. Lane 215: “able to drastically stimulate translation of both reporters.” There is not much of an increase in the Firefly signal, also there are no error bars, so we don’t know anything about the statistical significance.

We agree with the reviewer’s criticism that figure 4a is in its present form does not clearly show weather the PAM2w of Larp4b is important for mRNA stability or translation and decided to remove it entirely.

3) The authors should not refer to old PAM2 consensus sequences and use old numbering when referring to their sequence.  Although they cite Xie et al, (ref 23) which illustrates an updated consensus, they don't use it.  They erroneously refer to the conserved phenylalanine as position 8, replaced by tryptophan because they haven't rewritten this old manuscript to bring it up to date; in modern time nomenclature consensus, this is position 10 (see Xie et al, ref 23)

We have updated the consensus sequence numbering scheme.

4) The authors argue in the discussion that the structure that shows the downstream sequence beyond the PAM2 that engages the second MLLE domain in a noncanonical way may be meaningful but don't present any data to support that interpretation.  But nor do they discuss the data that argues against it, including from their data in fig 1C in which they concluded (line 131): "the wild-type protein formed a stoichiometric complex with PABPC1".

We have now made clear in the discussion that there is currently no evidence supporting a higher order stoichiometry of the PAM2w MLLE binding or cooperative effects from in vitro effects. We have toned down our assumption that such a stoichiometry would be sterically possible in vivo on PABC-bound poly(A) tails:

“There is currently no data that supports this behaviour in vitro in solution. Nevertheless, one could hypothesize that in vivo the PAM2w site located within the largely unstructured and flexible N-terminus of LARP4B could simultaneously contact two or more distinct MLLE units of neighbouring, poly(A) bound PABC molecules. “

A recent paper by Deragon shows alignment of the PAM2w sequences of multiple mammalian and other vertebrate LARP4B proteins (Deragon RNA Biology 2020, supplementary figure S3D). https://www.tandfonline.com/doi/full/10.1080/15476286.2020.1739930  The authors should consider/address that sequence homology is very high in the PAM2w sequence but is lower beyond it.

Thus, in discussion they should also consider that the second MLLE in the crystal is an artifact and discuss experiments using LARP4B constructs mutagenized at residues downstream of the PAM2w that are observed to be in contact with the second MLLE.

We agree with the critical view of the reviewer and have added the following sentence including the proposed literature reference in the Discussion section:

However, it should not go unnoticed that the cross-species sequence conservation is only high in the canonical part of the LARP4B PAM2w motif and comparably low in its variable part (Deragon 2020).

Finally, when first described in Results (line 150) the authors write: "Within the crystal structure, a single LARP4B-vPAM2 peptide and three neighboring MLLE molecules contact each other, producing an extended interface (Fig. 2B)." but there are only two MLLE domains shown in the figure 2 (there is no Fig. 2B).

We apologize for the mistake which we have corrected as follows:

Within the crystal structure, a single LARP4B-PAM2w complex contacts a neighbouring, symmetry related MLLE entity, providing an extended interface that creates a deep cleft in the interface region into which the peptide binds, running over or in between the surfaces of both symmetry-related MLLE molecules (Fig. 2).

5) Figure 4B: the IF pictures are not high quality, especially of the C-terminal fragment. If they have images that show more cells they should use those.  There are only 3 stress granules in the cells that overexpress the C-terminal fragment.

      A conceptual problem is that as presented, the authors argue that presence in stress granules, reflects translation, but this is incorrect.  According to current understanding, stress granules is where mRNAs reside that are not actively translated.

      The negative data here don't add anything to this paper, similar to the luciferase data.

We are unfortunately not able to repeat the experiments in the given time frame. In our opinion the quality of the pictures is sufficient to show the accumulation of Larp4b in stress granules upon arsenite treatment. We agree with the reviewer’s criticism in that the stress granule experiments do not allow us to draw conclusions regarding the role of the W63K mutation in translation. Nevertheless, we believe that these results further strengthen our overall conclusion that the W63K mutation alone is not able to disrupt LARP4B’s function in vivo. We decided therefore to tone down our conclusions to better reflect the indirect nature of our experimental approach.

Specific points:

Figure 1D: Two bands are present for PABPC1 in both input lanes and lane 3 corresponding to IP with WT LARP4B, but only the upper band was IP with LARP4B-W63K. The authors should comment on this.

We agree with the reviewer, but we do not have any experimental evidence on the influence of other factors or posttranslational modifications on LARP4B-W63K binding to PABPC1.

      How were the results quantified? Was this done with ECL or fluorescent antibodies? Was the HA-PABPC1 signal divided by the HA-LARP4B signal?

Figure 1C and D: Please indicate in the figures which lane is WT LARP4B and which lane is mutant LARP4B.
Figure 1C: How were the results quantified? Was the HA-PABPC1 signal divided by the HA-LARP4B signal?

The results were quantified with ECL and the HA-PABPC1 signal divided by the HA-LARP4B. Nonetheless we decided to remove the quantification and to tone down our conclusions to avoid confusion. We updated the figure legend in figure 1c.

Page 6, line158: “like in all other PAM2-MLLE complex structures reported so far.” This same observation is already reported for LARP4 PAM2w

We have removed the halve sentence claiming exclusivity for LARP4B.

Page 6, line 159: “Therefore, the generic pattern for the canonical PAM2 motif LNxxAxxF has to be extended to LNxxAxx[F/W]. “ Even though the authors cite Xie et al (2014), in which an updated consensus and logo is published that includes the F/W, they completely neglect to mention this.

We have replaced the above sentence by:

This behaviour conforms to the MLLE conserved binding sequence pattern reported previously (Xie, Kozlov, and Gehring 2014)

Line 195: "Thus, the interaction of vPAM2 with MLLE is not essential for LARP4B’s stimulatory role in translation." This must be deleted because they didn't measure translation, not by luciferase output as above and not by stress granules where translation doesn't occur

We agree and have deleted the sentence.

Line 232: “As the vPAM2-MLLE interaction of LARP4B with PABPC1 appeared to be non-essential for LARP4Bs effect on translation, we assumed, that a truncation of LARP4B lacking the N-terminal part of the protein (including the vPAM2 motif) should be sufficient to accumulate in stress granules." This rationale doesn't make sense, it must be deleted , see above
Line 251: “A comparison of all relevant atomic structures revealed that the major determinants or binding of vPAM2 and PAM2 motifs reside within their canonical part, i.e. in the parts that are shared between both motifs.” not all relevant atomic structures were used. What about LARP4?

We agree and have modified the sentence accordingly.

Minor:

In their description of fig 1 they show that mutation of the Trp (W) decreases binding to PABPC1 in vitro and less so in vivo by IP. They note (line 139): "These results suggest that the vPAM2 domain contributes to the interaction of LARP4B to PABPC1 in vitro and in vivo although in the latter case additional factors are likely to influence binding (see also discussion)." First, they do not address this in the discussion but more importantly, they should refer to the other region depicted in figure 1a rather than "additional factors".

We agree with the reviewer and added modified the sentence accordingly

Page 2, line 44” Moreover these cytoplasmic LARP proteins all harbor PAM2 motifs and likely bind PABPC1 [45-48]
human LARP6 does not have PAM2
Page 3, line 90: delete: , respectively

lines 151, 157: refer to fig 2A, 2B but there is only fig. 2.

We apologize for the mistake and have corrected the reference to Fig. 2.

Reviewer 2 Report

The authors present data for a variant PAM2 motif in LARP4B.  The structure appears to be of high quality and there are accompanying experiments that are helpful to describe the importance of this motif in LARP4B-associated promotion of translation.

The authors should make earlier reference to the variant PAM2 motif previously described for LARP4 (Yang et al., 2011) in their Results section, and should refer to the variant PAM2 motif in their structure as it is described in that paper (PAM2w) unless they can demonstrate how the variant PAM2 motif in their structure is substantially different. The paper would be most helpful in comparing/contrasting observations made in their structure and the one from Yang et. al. The description of the contacts to the more C-terminal section of the PAM2w to the second MLLE, for example, is more novel. Even if the canonical contacts are largely similar it would be good to note this and reinforce that both proteins bind PAM2w motifs similarly.

Specific comments:

Figure 1C: The difference between lanes 1 and 3 is not clear from the figure or the figure legend. Is lane 3 the mutant?

Figure 1C & D: Are there error bars or statistics for these histograms? If not it would be best to simply include the western blots.

Figure 2: Is it expected that the contacts between the C-terminal end of the PAM2 and MLLE B might be physiologically relevant? From the data presentation it is not clear whether these could be made to the same MLLE as the N-terminus of the PAM2 (MLLE A) or whether these would have to be made in the context of a dimer.  Further discussion of these results would add to the novelty of this work relative to Yang et al., 2011.

Figure 2B: The text makes reference to Figure 2B but it is missing.

Page 7, Line 185: Should this be “The N-terminal, canonical parts….” (not C-terminal?)

Figure 4: There are many issues with this figure. What is the difference between the two reporter constructs (firefly and renilla)? Are they both cap-dependent? Is it anticipated why renilla would be stimulated more than firefly? The difference between the left and right panels of Figure 4A are not obvious. Is the right hand panel the RNA levels, and it is from this that the authors indicate the observed effects are at the level of translation? This is not related in the figure, the figure legend or the text. If so then the right panel should not be measured as “luciferase percentage”.   Furthermore the figure legend is very confusing, saying that LARP4B is on the left and the mutant is on the right, yet both panels have both the wild-type and mutant. Further the figure legend says that HA-LARP4B luciferase was assigned as “100%” (line 223) while from the figure this appears to be set to what I presume is the empty vector (pHA?).

Author Response

Reviewer 2:

Open Review
(x) I would not like to sign my review report
( ) I would like to sign my review report
English language and style
( ) Extensive editing of English language and style required
( ) Moderate English changes required
(x) English language and style are fine/minor spell check required
( ) I don't feel qualified to judge about the English language and style
Yes        Can be improved        Must be improved        Not applicable
Does the introduction provide sufficient background and include all relevant references?
( )        (x)        ( )        ( )
Is the research design appropriate?
(x)        ( )        ( )        ( )
Are the methods adequately described?
( )        (x)        ( )        ( )
Are the results clearly presented?
( )        ( )        (x)        ( )
Are the conclusions supported by the results?
(x)        ( )        ( )        ( )
Comments and Suggestions for Authors
The authors present data for a variant PAM2 motif in LARP4B.  The structure appears to be of high quality and there are accompanying experiments that are helpful to describe the importance of this motif in LARP4B-associated promotion of translation.

The authors should make earlier reference to the variant PAM2 motif previously described for LARP4 (Yang et al., 2011) in their Results section, and should refer to the variant PAM2 motif in their structure as it is described in that paper (PAM2w) unless they can demonstrate how the variant PAM2 motif in their structure is substantially different. The paper would be most helpful in comparing/contrasting observations made in their structure and the one from Yang et. al. The description of the contacts to the more C-terminal section of the PAM2w to the second MLLE, for example, is more novel. Even if the canonical contacts are largely similar it would be good to note this and reinforce that both proteins bind PAM2w motifs similarly.

We now present the LARP4 PAM2w-MLLE complex by Yang et al. (3PKN) in Fig. 3. We also have given attention to the 3PKN structure in the results section, in particular with regard to the binding conformation of Trp22 the ‘variant’ W residue, differing in conformation from the LARP4 PAM2w Trp63:

“Judged from the frequency of occurrence within so far identified PAM2 motifs, pocket B has a preference for Phe, but is also able to harbour a Trp residue, as shown previously by Yang et al. (Yang et al. 2011) for Larp4 as well as for the Larp 4B PAM2w in the current study. Of note, the versatility of this pocket is also reflected by the different binding modes of LARP4(Trp22) and LARP4B(Trp63), see Fig.3.”

Specific comments:

Figure 1C: The difference between lanes 1 and 3 is not clear from the figure or the figure legend. Is lane 3 the mutant?

Figure 1C & D: Are there error bars or statistics for these histograms? If not it would be best to simply include the western blots.

We agree with the criticism and modified the figure accordingly. (See also response to reviewer 1)

Figure 2: Is it expected that the contacts between the C-terminal end of the PAM2 and MLLE B might be physiologically relevant? From the data presentation it is not clear whether these could be made to the same MLLE as the N-terminus of the PAM2 (MLLE A) or whether these would have to be made in the context of a dimer.  Further discussion of these results would add to the novelty of this work relative to Yang et al., 2011.

This is related to comment 4 from Referee 1. We fully agree with the critical view of the referees towards the physiological relevance of the observed peptide contacts to the symmetry related instance of the MLLE domain. We have now made clear in the discussion that there is currently no evidence supporting a higher order stoichiometry of the PAM2w MLLE binding or cooperative effects from in vitro effects. We have toned down our assumption that such a stoichiometry would be sterically possible in vivo on PABC-bound poly(A) tails:

“There is currently no data that supports this behaviour in vitro in solution. Nevertheless, one could hypothesize that in vivo the PAM2w site located within the largely unstructured and flexible N-terminus of LARP4B could simultaneously contact two or more distinct MLLE units of neighbouring, poly(A) bound PABC molecules. “

Figure 2B: The text makes reference to Figure 2B but it is missing.

We apologize for the mistake which we have corrected as follows:

Within the crystal structure, a single LARP4B-PAM2w complex contacts a neighbouring, symmetry related MLLE entity, providing an extended interface that creates a deep cleft in the interface region into which the peptide binds, running over or in between the surfaces of both symmetry-related MLLE molecules (Fig. 2).

Page 7, Line 185: Should this be “The N-terminal, canonical parts….” (not C-terminal?)

Figure 4: There are many issues with this figure. What is the difference between the two reporter constructs (firefly and renilla)? Are they both cap-dependent? Is it anticipated why renilla would be stimulated more than firefly? The difference between the left and right panels of Figure 4A are not obvious. Is the right hand panel the RNA levels, and it is from this that the authors indicate the observed effects are at the level of translation? This is not related in the figure, the figure legend or the text. If so then the right panel should not be measured as “luciferase percentage”.   Furthermore the figure legend is very confusing, saying that LARP4B is on the left and the mutant is on the right, yet both panels have both the wild-type and mutant. Further the figure legend says that HA-LARP4B luciferase was assigned as “100%” (line 223) while from the figure this appears to be set to what I presume is the empty vector (pHA?).

We agree with the criticism and removed the figure accordingly. (See also response to reviewer 1)

Reviewer 3 Report

In this manuscript the authors provide a structure of the MLLE domain of PABPC1 with a peptide from the LARP4B protein. Important for these protein families members, the interaction uses a non-canonical interface that extends the amino acid residues that are capable of participating in these interactions. Generally, the data is clearly presented, of good quality, and fairly discussed. Prior to publication, I would suggest the following issues be addressed: 

  1. The intro is short and general, and for a non-expert, the background information is not sufficient to allow the reader to understand why these interactions are important and worth understanding. 
  2. There are no indications that the data in Figure 1 has been repeated and what variability there is in these assays (i.e. quantitation of the data). This is important because the reported changes in the in vivo pulldown are subtle, plus data in figure 4 would suggest that the change is not impacting the biology. Shorter exposures for the blots in 1D should also be presented since it appears they may be multiple bands. 
  3. In figure 4A, if the change in the interaction is slight (as suggested in figure 1), could the high level of expression and time point lead to seeing no effect on translation. In other words, what is the dynamic range of the system across different levels of LARP4B expression or time? The impact of the RRM-Ct fragment in the translation assay and that of a mutant known to alter LARP4B activity should also be included for comparison. The left graph in figure 4A has no indication of error bars / replicates, these should be added.
  4. Figure 4 B should include quantification of the # foci in each channel and the # that overlap in multiple experiments, as right now only a single cell is shown as an example for each condition. 
  5. The identification of a variant motif is reported, these results would be more impactful if such motifs were reported/common in other proteins. Does this variant motif appear in other proteins known to interact with the PABP family?

Author Response

Reviewer 3:

Open Review
(x) I would not like to sign my review report
( ) I would like to sign my review report
English language and style
( ) Extensive editing of English language and style required
( ) Moderate English changes required
(x) English language and style are fine/minor spell check required
( ) I don't feel qualified to judge about the English language and style
Yes        Can be improved        Must be improved        Not applicable
Does the introduction provide sufficient background and include all relevant references?
( )        (x)        ( )        ( )
Is the research design appropriate?
(x)        ( )        ( )        ( )
Are the methods adequately described?
(x)        ( )        ( )        ( )
Are the results clearly presented?
(x)        ( )        ( )        ( )
Are the conclusions supported by the results?
(x)        ( )        ( )        ( )
Comments and Suggestions for Authors
In this manuscript the authors provide a structure of the MLLE domain of PABPC1 with a peptide from the LARP4B protein. Important for these protein families members, the interaction uses a non-canonical interface that extends the amino acid residues that are capable of participating in these interactions. Generally, the data is clearly presented, of good quality, and fairly discussed. Prior to publication, I would suggest the following issues be addressed:

The intro is short and general, and for a non-expert, the background information is not sufficient to allow the reader to understand why these interactions are important and worth understanding.
There are no indications that the data in Figure 1 has been repeated and what variability there is in these assays (i.e. quantitation of the data). This is important because the reported changes in the in vivo pulldown are subtle, plus data in figure 4 would suggest that the change is not impacting the biology. Shorter exposures for the blots in 1D should also be presented since it appears they may be multiple bands.

We agree with the criticism and modified the figure 1C and D accordingly. (See also response to reviewer 1)

In figure 4A, if the change in the interaction is slight (as suggested in figure 1), could the high level of expression and time point lead to seeing no effect on translation. In other words, what is the dynamic range of the system across different levels of LARP4B expression or time? The impact of the RRM-Ct fragment in the translation assay and that of a mutant known to alter LARP4B activity should also be included for comparison. The left graph in figure 4A has no indication of error bars / replicates, these should be added.
Figure 4 B should include quantification of the # foci in each channel and the # that overlap in multiple experiments, as right now only a single cell is shown as an example for each condition.

We agree with the criticism and removed figure 4A. (See also response to reviewer 1)

The identification of a variant motif is reported, these results would be more impactful if such motifs were reported/common in other proteins. Does this variant motif appear in other proteins known to interact with the PABP family?

This point is related to the criticism of the other referees to refer to the existing and related LARP4A-MLLE crystal structure. We have now included these references where appropriate and updated also Fig. 3 as well as the presently common motif name “PAM2w”.

Reviewer 4 Report

The article by Grimm and coworkers investigates the interaction between the MLLE domain of the cytoplasmic PABP and LARP4B. They found a variant PAM2 motif in LARP4B (vPAM2) responsible for this association and reported the crystal structure of the complex between the MLLE domain and a vPAM2 peptide. In cell investigations showed that this interaction is not essential for LARP4B role in translation.

This is a well-conducted and rigorous study showing interesting and important results. There are nonetheless major points that need to be addressed before publication, as listed below.

Major points:

  1. The manuscript has been written overlooking the fact that the closely related LARP4A (or LARP4) protein also contains a very similar variant PAM2 motif, which was extensively characterized structurally and functionally in 2011 (Yang at al, La-related protein 4 binds poly(A), interacts with the poly(A)-binding protein MLLE domain via a variant PAM2w motif, and can promote mRNA stability. Mol Cell Biol. 2011, 31(3):542-56. PMID:21098120). As already found in LARP4A, in LARP4B the normally invariant phenylalanine has been replaced by a tryptophan, albeit the sequence differs downstream of this residue (an important difference in what the authors describe a ‘variable PAM2 part’). The manuscript in its entirety has to be edited keeping these considerations in mind and in the context of relevant available literature, from introduction, to results, discussion and figures. The last sentence (page 10-11, lines 279-285) has to be taken out and the entire manuscript re-worked appropriately. Sentences throughout the manuscript and figures seem to allude that this was the first case showing the F/W substitution: these need to be changed. Moreover, a thorough comparison of the 2 variant PAM2 motifs from LARP4A and LARP4B, both from a structural and a functional point of view is mandatory and will considerably strengthen the paper. It was quite surprising for example that LARP4A PAM2w was deliberately left out from figure 3 for example. A first structural comparison was discussed in a recent review (Dock-Bregeon at al, RNA biology 2019, doi: 10.1080/15476286.2019) which could be a starting point from a further elaboration here, especially focusing on the differences in the PAM2 ‘variable part’ that the authors have nicely highlighted. A functional comparison is also expected of both vPAM2 from LARP4A and LARP4B.

  1. The finding that LARP4B vPAM2 mutant W63K still localizes to stress granules has been devised from the assumption that such localization is PABP-interaction dependent. In my opinion this may be an overinterpretation: other proteins localize to stress granules using other mechanisms (rather than direct association with PABP), and such a possibility needs to be considered here.

Author Response

Reviewer 4:

Open Review
(x) I would not like to sign my review report
( ) I would like to sign my review report
English language and style
( ) Extensive editing of English language and style required
( ) Moderate English changes required
(x) English language and style are fine/minor spell check required
( ) I don't feel qualified to judge about the English language and style
Yes        Can be improved        Must be improved        Not applicable
Does the introduction provide sufficient background and include all relevant references?
( )        ( )        (x)        ( )
Is the research design appropriate?
(x)        ( )        ( )        ( )
Are the methods adequately described?
(x)        ( )        ( )        ( )
Are the results clearly presented?
( )        ( )        (x)        ( )
Are the conclusions supported by the results?
( )        ( )        (x)        ( )
Comments and Suggestions for Authors
The article by Grimm and coworkers investigates the interaction between the MLLE domain of the cytoplasmic PABP and LARP4B. They found a variant PAM2 motif in LARP4B (vPAM2) responsible for this association and reported the crystal structure of the complex between the MLLE domain and a vPAM2 peptide. In cell investigations showed that this interaction is not essential for LARP4B role in translation.

This is a well-conducted and rigorous study showing interesting and important results. There are nonetheless major points that need to be addressed before publication, as listed below.

Major points:

The manuscript has been written overlooking the fact that the closely related LARP4A (or LARP4) protein also contains a very similar variant PAM2 motif, which was extensively characterized structurally and functionally in 2011 (Yang at al, La-related protein 4 binds poly(A), interacts with the poly(A)-binding protein MLLE domain via a variant PAM2w motif, and can promote mRNA stability. Mol Cell Biol. 2011, 31(3):542-56. PMID:21098120). As already found in LARP4A, in LARP4B the normally invariant phenylalanine has been replaced by a tryptophan, albeit the sequence differs downstream of this residue (an important difference in what the authors describe a ‘variable PAM2 part’). The manuscript in its entirety has to be edited keeping these considerations in mind and in the context of relevant available literature, from introduction, to results, discussion and figures. The last sentence (page 10-11, lines 279-285) has to be taken out and the entire manuscript re-worked appropriately. Sentences throughout the manuscript and figures seem to allude that this was the first case showing the F/W substitution: these need to be changed. Moreover, a thorough comparison of the 2 variant PAM2 motifs from LARP4A and LARP4B, both from a structural and a functional point of view is mandatory and will considerably strengthen the paper. It was quite surprising for example that LARP4A PAM2w was deliberately left out from figure 3 for example. A first structural comparison was discussed in a recent review (Dock-Bregeon at al, RNA biology 2019, doi: 10.1080/15476286.2019) which could be a starting point from a further elaboration here, especially focusing on the differences in the PAM2 ‘variable part’ that the authors have nicely highlighted. A functional comparison is also expected of both vPAM2 from LARP4A and LARP4B.

We thank the Referee 4 for pointing this out in such detail. We have now renamed the variant LARP4/LARP4B PAM2 motif as PAM2w and included the Yang et al. LARP4 PAM2w structure in Fig. 3. We have also included an appropriate comparison to the LARP4 PAM2w complex in the results and discussion section. This is related to several points of the other Referees.

The finding that LARP4B vPAM2 mutant W63K still localizes to stress granules has been devised from the assumption that such localization is PABP-interaction dependent. In my opinion this may be an overinterpretation: other proteins localize to stress granules using other mechanisms (rather than direct association with PABP), and such a possibility needs to be considered here.

We agree with the reviewer and have modified our conclusions accordingly. We now only state that the uncovered interaction seems to not be essential for LARP4B’s interaction to PABPC1 or its target mRNAs.

Round 2

Reviewer 1 Report

The major criticism of the first version of this manuscript by Referee #1 included the following: ”Just because the authors describe a structure from a decade ago doesn't mean that they should resubmit a paper that was mostly written at that time, as this paper is distastefully presented. The authors should re-write the manuscript in the here-and-now, not just inserting citations of current literature, but incorporating the science into proper context and interpretation of the results.” The critique by Referee #4 stated; “Sentences throughout the manuscript and figures seem to allude that this was the first case showing the F/W substitution: these need to be changed.” Another criticism was that the authors should use modern nomenclature when referring to the PAM2 consensus, especially position 10 when referring to the F/W.  An authors response to a criticism was that they “included an appropriate comparison to the LARP4 PAM2w complex in the results and discussion section.”  However, although the opening paragraph of the Results section was edited to refer to W as position 10 as suggested, it was otherwise left unedited to leave the impression that the authors were the first to document W in place of F in a PAM2.  However, this is in fact proven to be wrong/incorrect by the very document that they hope to publish in this manuscript, it is also intentionally misleading to this review process, to the readers and to history. The first paragraph of version 2 ends “Hence, we assumed that the sequence in LARP4B represents a variant PAM2 motif (which we termed PAM2w)” and this is obviously unscholarly and noncollegial.  The logic of this manuscript and historical events document it to be untrue because 1) LARP4 is included in their Fig 4B alignment, 2) Bayfield et al reported in Feb 2010 online doi:10.1016/j.bbagrm.2010.01.011 graphically and in text, “the presence of a highly conserved PAM2 in LARP4 and 4B family members,” and 3) it was established by Yang et al online Nov 2010 that LARP4 is the pioneer PAM2w.  Further, their work does not document a PAM2w in LARP4B because they have shown that mutation of W10 has no effect on any biological activity of LARP4B. By contrast, Yang et al documented in 2011 that mutation of PAM2w of LARPw by deletion or substitution disrupted association with polysomes. The authors should replace the outdated Albrecht 2004, and Kozlov 2010 citations in their first paragraph of Results section, with e.g., Xie et al 2014, and not cryptically try to rewrite the history of documentation of the PAM2w.

In their response, the authors claim to have fixed reference to fig 2a and 2b because there is no a and b parts of figure 2, but they have not changed the text: “2A shows the molecular details of the peptide/protein interaction while figure 2B provides an overview over the arrangement within the crystal lattice.” There is no fig 2B that shows details of crystal lattice.

A critcism of the first version was that the authors referred to three MLLE molecules in the crystal even though their figures showed only two molecules.  On the bottom of p10, version 2, they refer to three distinct MLLE moleucles.  

The penultimate paragraph of the Discussion  “It was hence not entirely unexpected that a missense mutation in PAM2w or even the entire deletion of the motif failed to interfere with the recruitment of LARP4B to this mRNP. We speculate that the interface between PAM2w and MLLE contributes only partially to the stable recruitment of LARP4B”  This discussion bizarrely fails to mention the previously characterized C-terminal region of LARP4B depicted in Fig1 that interacts with PABP and instead refer to other cellular factors that might be responsible for recruiting LARP4B to PABP despite disrupting the PAM2w. 

Author Response

The major criticism of the first version of this manuscript by Referee #1 included the following: ”Just because the authors describe a structure from a decade ago doesn't mean that they should resubmit a paper that was mostly written at that time, as this paper is distastefully presented. The authors should re-write the manuscript in the here-and-now, not just inserting citations of current literature, but incorporating the science into proper context and interpretation of the results.” The critique by Referee #4 stated; “Sentences throughout the manuscript and figures seem to allude that this was the first case showing the F/W substitution: these need to be changed.” Another criticism was that the authors should use modern nomenclature when referring to the PAM2 consensus, especially position 10 when referring to the F/W.  An authors response to a criticism was that they “included an appropriate comparison to the LARP4 PAM2w complex in the results and discussion section.”  However, although the opening paragraph of the Results section was edited to refer to W as position 10 as suggested, it was otherwise left unedited to leave the impression that the authors were the first to document W in place of F in a PAM2.  However, this is in fact proven to be wrong/incorrect by the very document that they hope to publish in this manuscript, it is also intentionally misleading to this review process, to the readers and to history. The first paragraph of version 2 ends “Hence, we assumed that the sequence in LARP4B represents a variant PAM2 motif (which we termed PAM2w)” and this is obviously unscholarly and noncollegial. 

We agree with the referee’s criticism and apologize for having this overlooked in the last revision. It was never our intention to mislead the reader. We have now included appropriate references here and changed the passage to:

“The sequence in LARP4B represents a variant PAM2 motif termed PAM2w xxx (Dock-Bregeon, Lewis, and Conte 2019; Yang et al. 2011) that can interact with the MLLE domain of PABC1. To test this for LARP4B, we co-translated […]”

The logic of this manuscript and historical events document it to be untrue because 1) LARP4 is included in their Fig 4B alignment, 2) Bayfield et al reported in Feb 2010 online doi:10.1016/j.bbagrm.2010.01.011 graphically and in text, “the presence of a highly conserved PAM2 in LARP4 and 4B family members,” and 3) it was established by Yang et al online Nov 2010 that LARP4 is the pioneer PAM2w.  Further, their work does not document a PAM2w in LARP4B because they have shown that mutation of W10 has no effect on any biological activity of LARP4B. By contrast, Yang et al documented in 2011 that mutation of PAM2w of LARPw by deletion or substitution disrupted association with polysomes. The authors should replace the outdated Albrecht 2004, and Kozlov 2010 citations in their first paragraph of Results section, with e.g., Xie et al 2014, and not cryptically try to rewrite the history of documentation of the PAM2w.

We want to emphasize again that we fully agree with the referee’s sight of the historical developments in the LARP field and that it was never our intention to be deceptive in any way. We therefore apologize again for the mistake and have exchanged the citations as proposed.

In their response, the authors claim to have fixed reference to fig 2a and 2b because there is no a and b parts of figure 2, but they have not changed the text: “2A shows the molecular details of the peptide/protein interaction while figure 2B provides an overview over the arrangement within the crystal lattice.” There is no fig 2B that shows details of crystal lattice.

We have corrected this mistake.

A critcism of the first version was that the authors referred to three MLLE molecules in the crystal even though their figures showed only two molecules.  On the bottom of p10, version 2, they refer to three distinct MLLE moleucles.

It is undoubtful that the PAM2w peptide in fact contacts three neighboring MLLE molecules in the crystal lattice.  However, we agree that – based on the current available data- no physiological implications can be derived from this observation. Within this passage, we correctly refer to the implications for the in vitro crystallization behavior.

The penultimate paragraph of the Discussion  “It was hence not entirely unexpected that a missense mutation in PAM2w or even the entire deletion of the motif failed to interfere with the recruitment of LARP4B to this mRNP. We speculate that the interface between PAM2w and MLLE contributes only partially to the stable recruitment of LARP4B”  This discussion bizarrely fails to mention the previously characterized C-terminal region of LARP4B depicted in Fig1 that interacts with PABP and instead refer to other cellular factors that might be responsible for recruiting LARP4B to PABP despite disrupting the PAM2w.

We agree with the referee and apologize for our oversight. We added an additional sentence to the discussion: “Most likely recruitment of LARP4B to its mRNPs is aided by its the C-terminal region which was shown  to co-immunoprecipitate with PABPC1 independently of the PAM2w (Schaffler et al. 2010).”

Reviewer 3 Report

As indicated and asked for in the first review of this paper, there are no indications that the data in Figure 1 has been repeated and what variability there is in these assays (i.e. quantitation of the data). This is important because I do not agree that there are changes in the in vivo pulldown. Figure 4 also supports the idea that this residue is not important in vivo for interactions driving protein localization in stress granules. If the authors want to make the claim that the interaction is changed in Fig. 1D it should be quantitated properly. Again as requested, a shorter exposure of the gel in 1D should also be presented since it appears they may be multiple bands.

Related to above, images of a single cell are not adequate without supporting quantitation. This is why I requested that Figure 4B  include quantification of the # foci in each channel and the # that overlap in multiple experiments. This is also important because the foci in the red channel are very hard to see. Improving the quality of the presented images is needed.

Finally, reviewer 1 has pointed out the existence of this PAM2w motif in another protein based on published data, yet the authors still state in the paragraph starting on line 133 of the results that they discovered this motif and "assumed that the sequence in LARP4B represents a variant PAM2 motif (which we termed PAM2w)". Given this prior research, I am not sure why they assumed this was a variant or how the named this motif PAM2w when it was already called this in the literature. The text needs to be written to present the data in the context of current literature and to fairly represent what they found vs. others. 

Author Response

As indicated and asked for in the first review of this paper, there are no indications that the data in Figure 1 has been repeated and what variability there is in these assays (i.e. quantitation of the data). This is important because I do not agree that there are changes in the in vivo pulldown. Figure 4 also supports the idea that this residue is not important in vivo for interactions driving protein localization in stress granules. If the authors want to make the claim that the interaction is changed in Fig. 1D it should be quantitated properly. Again as requested, a shorter exposure of the gel in 1D should also be presented since it appears they may be multiple bands.

Related to above, images of a single cell are not adequate without supporting quantitation. This is why I requested that Figure 4B  include quantification of the # foci in each channel and the # that overlap in multiple experiments. This is also important because the foci in the red channel are very hard to see. Improving the quality of the presented images is needed.

We fully agree with the referee that both the pulldown experiments in figure 1b and c as well as the immunofluorescence images in figure 4B would benefit from quantification of independent biological replicates. Unfortunately, due to the shut down of several of our facilities as well as the home office of most of the co-authors of this study due to the Coronavirus regulations we are unable to perform the requested experiments and analysis in the requested time frame. We apologize to the reviewers and the editor for these complications under the current situation. As a consequence we further toned down our conclusions both in the results section:

“Mutant LARP4B bound slightly weaker to PABPC1 as compared to the wild type. However, this effect seemed to not be as strong as in vitro. These results gave us a first hint that the PAM2w domain might contribute to the interaction of LARP4B to PABPC1 in vitro but that in vivo additional factors or other regions in LARP4B could likely influence binding.”

And

“while strongly affecting LARP4B’s interaction to PABPC1 in vitro seems to not be sufficient in vivo to disrupt the recruitment of LARP4B to stress granules. These results suggest that additional until now unknown factors contribute the recruitment of LARP4B to its native mRNPs.”

However, if this is deemed insufficient, we would leave the decision to remove the immunofluorescence image to the editor.

Finally, reviewer 1 has pointed out the existence of this PAM2w motif in another protein based on published data, yet the authors still state in the paragraph starting on line 133 of the results that they discovered this motif and "assumed that the sequence in LARP4B represents a variant PAM2 motif (which we termed PAM2w)". Given this prior research, I am not sure why they assumed this was a variant or how the named this motif PAM2w when it was already called this in the literature. The text needs to be written to present the data in the context of current literature and to fairly represent what they found vs. others.

We agree with the referee’s criticism and apologize for having this overlooked in the last revision (this is related to a similar comment made by referee 1). We have now included appropriate references here and changed the passage to:

“The sequence in LARP4B represents a variant PAM2 motif termed PAM2w xxx (Dock-Bregeon, Lewis, and Conte 2019; Yang et al. 2011) that can interact with the MLLE domain of PABC1. To test this for LARP4B, we co-translated […]”

Reviewer 4 Report

The revised article by Grimm and coworkers is much improved from the earlier version. The authors did largely address the points raised, albeit not completely.

Minor points:

  1. The manuscript does now acknowledge earlier published data on the variant PAM 2 motif from LARP4A. Nonetheless a few paragraphs, indicated below, have not been modified accordingly, and I would recommend addressing this before publication.

- Lines 137-142. This section now make less sense as the concept has been introduced in the introduction, and it still appear to contain elements of the old narrative that the variant PAM 2 motif was firstly discovered here. This needs to be modified accordingly

- Line 144 –‘PAM2w’ has not been termed by the authors of this manuscript – please remove the ‘we’ and acknowledge the first source.

  1. In this new version of the manuscript a comparison between the PAM2w motif of LARP4A and LARP4B has been included. This is a significant improvement of the manuscript. However, there is a claim that in my opinion is not substantiated, that the actual variant W residue (W22 in LARP4A and W63 in LARP4B) have different binding modes (lines 78-80 and line 239). There is small change in the conformation of the W in the bound state with the MLLE domain, but in my opinion this seems small and probably insignificant, perhaps a result of a crystal packing? Could the authors can actually support these statements with adequate explanations, expand on exactly what is different and why these may be significant? And if not, these statements need to be taken out or modified substantially.

3. There seems to be a lot of inconsistencies in the nomenclature used, e.g. LARP4 or LARP4A, LARP vs Larp etc.

Author Response

The manuscript does now acknowledge earlier published data on the variant PAM 2 motif from LARP4A. Nonetheless a few paragraphs, indicated below, have not been modified accordingly, and I would recommend addressing this before publication.
- Lines 137-142. This section now make less sense as the concept has been introduced in the introduction, and it still appear to contain elements of the old narrative that the variant PAM 2 motif was firstly discovered here. This needs to be modified accordingly

- Line 144 –‘PAM2w’ has not been termed by the authors of this manuscript – please remove the ‘we’ and acknowledge the first source.

We did so.

In this new version of the manuscript a comparison between the PAM2w motif of LARP4A and LARP4B has been included. This is a significant improvement of the manuscript. However, there is a claim that in my opinion is not substantiated, that the actual variant W residue (W22 in LARP4A and W63 in LARP4B) have different binding modes (lines 78-80 and line 239). There is small change in the conformation of the W in the bound state with the MLLE domain, but in my opinion this seems small and probably insignificant, perhaps a result of a crystal packing? Could the authors can actually support these statements with adequate explanations, expand on exactly what is different and why these may be significant? And if not, these statements need to be taken out or modified substantially.

This refers to the different sidechain conformations of the Trp residue bound to the pocket. We have removed the term ‘binding mode’:

“Of note, the versatility of this pocket is also reflected by its ability to harbour the bound tryptophane in different sidechain conformations (compare LARP4(Trp22) and LARP4B(Trp63) in Fig. 3).”

There seems to be a lot of inconsistencies in the nomenclature used, e.g. LARP4 or LARP4A, LARP vs Larp etc.

We thank the referee for pointing this out and have corrected the inconsistencies.